# Novel use of an adapted UV Double Monochromator for measurements of global and direct irradiance, ozone and aerosol

Alexander Geddes[1], Ben Liley[1], Richard McKenzie[1], Michael Kotkamp[1], Richard Querel[1]

[1] National Institute of Water & Atmospheric Research (NIWA), Lauder, New Zealand

*Correspondence to*: Alex Geddes alex.geddes@niwa.co.nz

**Abstract.** A novel ultraviolet spectrometer has been developed and tested over 10 years at Lauder, New Zealand. The system, UV2, makes alternating measurements of the global and direct UV irradiance and can therefore be used to measure ozone and aerosol optical depth. After an analysis of the stability of UV2, these measurements, along with UV irradiance are compared to relevant observations made by an additional UV spectrometer (UV4), a Dobson spectrophotometer (#72) and

two radiometers measuring aerosol optical depth - a Prede Skyradiometer and a Middleton Solar radiometer (SP02). UV2 irradiance is shown to be lower than UV4 by between 2.5-3.5%, with a standard deviation of a similar magnitude. Total column ozone values are shown to agree with Dobson values with a mean bias of 2.57 Dobson units (DU) and standard deviation of 1.15 DU when using the direct sun measurements. Aerosol optical depth at 400 nm - 412 nm and 500 nm agrees to within 0.015 and is comparable to the difference between the reference radiometers. Further work is needed, particularly

in the radiometric calibration at longer wavelengths, in order to determine if this instrument can supersede or enhance measurements made by the Dobson or the aerosol radiometers.

## 1 Introduction

Soon after anthropogenically induced threats to the ozone layer first manifested with the formation of the Antarctic ozone hole in the 1980s, a measurement programme was initiated at Lauder, New Zealand (45°S, 170°E, alt 370 m) to measure the

spectral distribution of UV irradiance at the surface. The aims of the programme were to characterise and understand its variability, and to monitor changes due to ozone depletion. Double monochromator systems were developed to measure the global spectral irradiance incident on a horizontal surface over the wavelength range from 290 nm to 450 nm to an accuracy of +/-5%. An early use of the data showed the inverse relationship between ozone and UV (McKenzie et al., 1991). Shortly afterwards the data showed that peak irradiances were unexpectedly greater than at similar latitudes in Europe (Seckmeyer &

McKenzie, 1992). Before the end of the same decade, the time series showed the increases in peak summertime UV at Lauder over the 1990s due to ozone depletion (McKenzie et al., 1999). The measurement systems as developed met the demanding criteria of the Network for the Detection of Stratospheric Change (NDSC, later NDACC). With continued refinement including extended range (to 285 nm), better analog-to-digital conversion, custom diffuser, fibre-optic coupling, and containerisation with integrated calibration, instruments were deployed at several other clean air sites around the world.

These spectrometer systems have since been used to quantify the effects of volcanic aerosols (Zeng et al., 1994), surface

albedo (McKenzie et al., 1996; McKenzie et al., 1998), altitude (McKenzie et al., 2001) and air pollution (McKenzie et al., 1994; McKenzie et al., 2008) on surface UV irradiance. They've also been used to find the relationship between irradiance and actinic flux (Hofzumahaus et al., 2004; McKenzie et al., 2002) in validation of satellite data (Tanskanen et al., 2007) and studies of vitamin D exposure (McKenzie et al., 2009), though the input geometry is not well suited to that task. A recent publication showed the effectiveness of the Montreal Protocol in curbing increases in UV due to ozone depletion at these sites and others since the turn of the century (McKenzie et al., 2019). Most recently, the decades of data from all the NDACC sites with these spectrometers have revisited the 1991 theme in closely characterising the relationship of UV irradiance to column ozone (McKenzie et al., 2022).

Over recent years, the focus of UV research at Lauder has broadened. Previously, it was to understand the effects of ozone depletion. Our new foci include understanding effects of other factors, in particular the effects of clouds and aerosols, which are likely subject to future changes, particularly in the event of possible climate interventions that would involve aerosol seeding of the upper atmosphere. We want to be able to quantify their effects on UV doses relevant to skin damage and to vitamin D production (i.e., both negative and positive health outcomes), and the effects on energy balance and climate change.

With those broader aims in mind, we developed a new instrument with two new capabilities: Firstly, the ability to measure irradiances over the wider wavelength range from 285 nm to 600 nm (to be supplemented with diode array spectrograph measurements for longer wavelengths extending into the IR region), and secondly to be able to measure direct radiance as well as global irradiances on a horizontal surface.

While there are several instruments available that have been adapted to make direct sun observations of UV irradiance and ozone , such as utilizing the BTS-2048-UV array spectroradimater as in (Zuber et al., 2018), Brewer spectrophotometers (Kerr et al., 1985) and Pandora spectrometers (Tzortziou et al., 2012), it was decided to build a system at Lauder that shares commonality with the existing range of Bentham spectrometers in order to retain the quality and heritage of the previous generation of instruments.

Here we demonstrate the capability of the new instrument, using data collected over several years of operation at Lauder. We describe this instrument in section 2 and the reference instruments in section 3. In section 4 we assess the instrument's performance over time with comparisons to stability and calibration lamps, and this is supported in section 5 where we show comparisons of the measured global UV. In section 6 we assess the ozone products derived from global and direct irradiances to those from a Dobson spectrophotometer. Lastly, we study measurements of the aerosol optical depth (AOD) from the direct irradiance (7) before summarising our findings in section 8.

## 2 NIWA UV2 Spectrometer

This paper focuses on the use of the "NIWA" Bentham double monochromator system named UV2. This is a standard NIWA UV system (Bernhard et al., 2008; Tanskanen et al., 2007; Wuttke et al., 2006) except that the long wavelength limit is extended from 450 nm to 600 nm, and it also incorporates two entrance ports instead of one. The standard port is connected via fibre-optic to a shaped PTFE diffuser designed to measured cosine weighted global irradiances, and the other port receives radiation from a fibre-optic connected to a telescope mounted on a solar tracker. By using a motor-driven

folding mirror, UV2 alternates between the two sources, the repeatability of which in the short term is better than 1%. Longer term changes caused by mechanical wear need to be accounted for and is done so through traceable calibration as discussed later. This is a novel approach which provides the fully automated measurements of global and direct irradiances, the combination of which, obtained within 15 minutes of each other, can be used to approximate the diffuse irradiance.

In line with our previous sampling strategy, the instrument is programmed to acquire global irradiance spectra at 15-minute

intervals over the midday period, and at 5-degree steps in solar zenith angle (SZA) for SZA <=95 outside the 2 hours closest to solar noon. These are interleaved with measurements of direct beam irradiances. Each spectrum takes approximately 4.5 minutes to complete, using a forward and reverse scan, with the timestamp at the turn around point. The use of a forward and reverse pair minimises the impact of the variation in solar zenith angle over the course of the measurement. From the resulting irradiance spectra, which have an average resolution of 0.65 nm (oversampled to 0.2 nm), several integrated

quantities are calculated. In this paper we principally use UVA and UVB irradiances which represent integrals over the wavelength ranges 315 nm - 400 nm and 280 nm - 315 nm, respectively. We also integrate the spectrum above 400 nm to assess the performance at longer wavelengths. To aid with data quality assurance and quality control (QAQC), particularly for cloud interference, a diode is mounted to the entrance aperture of the spectrometer, with its value monitored throughout the spectral measurement. Variations denote changes in the incoming radiation, primarily caused by clouds moving across

the sun.

## 3 Auxiliary Instrumentation

For direct comparison we use the collocated NIWA UV4 spectrometer, which is based on an Acton double monochromator and is the primary global UV irradiance measurement at Lauder (NDACC, (Wuttke et al., 2006). The global diffuser used by UV4 is a commercial Schreder heated diffuser. As a reference for ozone, we make use of the Dobson #072

spectrophotometer, which measures direct sun total column ozone (TCO). Dobson #072 is a NOAA instrument that has been operating at Lauder since 1986. Its zenith-sky measurements have been automated enabling regular Umkehr profile measurements. Only direct sun TCO are used in this study to enable like-for-like comparison. Dobson #072's processed data are submitted regularly to NDACC and WOUDC.

Lauder is, and has been since 1998, part of the Baseline Surface Radiation Network (BSRN, https://bsrn.awi.de/). As a component of the BSRN instrumentation, it hosts a 4 channel Middleton Solar SP02 radiometer mounted on an active solar tracker (Liley & Forgan, 2009). The system provides AOD observations calibrated by the Langley method at 412 nm, 500 nm, 610 nm, 778 nm as well as the tracking wavelength of 867 nm. This dataset has been filtered for periods of unobscured sun and for solar zenith angles less than 85 degrees (Alexandrov et al., 2004). Liley and Forgan also note that the AOD at

Lauder is some of the lowest in the world which presents a significant challenge to the accuracy of the Langley calibration. In this study, we use the AOD at 412 nm and 500 nm, the latter of which is corrected for Ozone.

Aerosol observations at Lauder are also made using a Prede Skyradiometer providing measurements of AOD aerosol single scattering albedo (SSA) and the aerosol Ångström exponent (alpha). The Prede POM-02 Skyradiometer is an active sun-

tracking radiometer with 11 wavebands from 340 nm to 2200 nm. It has been operating at the Lauder site since 2011 as part of the SKYNET aerosol network, and it is used in ground-based validation by NIES of GOSAT data products. At regular intervals the instrument breaks from sun tracking to scan the sky in steps from the horizon to the zenith and an almucantar from the solar azimuth through due north, plus less frequent gridded scans across the solar disk. The direct sun measurements are used to calculate AOD, SSA, and alpha, according to standard SKYNET protocols, at the Centre for

Environmental Remote Sensing (CEReS), Chiba University (Japan) and are available from http://atmos3.cr.chiba-u.jp/skynet/lauder/lauder.html.

**4 Stability and Calibration**

The medium-term stability of UV2 is assessed by monthly spectral measurements of a 45W quartz-halogen lamp, with

measurements being made in two modes: a constant lamp current mode and a constant intensity mode where a UVA diode is used in a feedback loop. These allow us to assess the radiometric stability of both the lamp and the spectrometer. Measurements are also undertaken using a low-pressure mercury lamp to monitor the stability of the line shape and wavelength alignment. Both lamp systems are contained in a calibration unit inside the spectrometer enclosure. During calibrations, the global and direct port heads are dismounted and placed inside the calibration unit. The 45W lamps can be

calibrated against FEL lamps and used as a transfer standard for systems that are operated at locations remote from calibration facilities (such as for our spectrometers systems operating in Alice Springs, Australia). However, the transfer introduces an additional uncertainty of 1% - 2%. Therefore, for instruments located at Lauder, such as UV2, absolute calibrations are made directly against 1000 W FEL lamps as follows.

Primary irradiance calibrations, traceable to National Institute of Standards and Technology (NIST) standards, are made using an independently calibrated 1000W Quartz Tungsten Halogen FEL Lamp (QTH), typically at 6-month intervals. These

calibrations allow the voltage response of the detectors to irradiance to be characterised as a function of wavelength. The calibrations are then applied by a time-weighted mean to real observations. The calibrations carry a radiometric uncertainty of 3%. Unfortunately, for budgetary reasons, during the first 5 years of operation these calibrations were much less frequent
(only 2 or 3 over that 5-year period). Fortunately, the programme of stability measurements was continued so that a record of performance was maintained. However, no corrective actions were taken.

Figure 1 shows the change in the measured UVB, UVA, and the integral for wavelengths over 400 nm for the global and direct ports using the 45W lamp in constant intensity mode. The constant current mode is used to assess the performance of
the 45W lamp and is shown for the global measurements. The data shown are processed assuming a fixed calibration (orange) to highlight instrumental drift, and then with the time-weighted mean calibration (blue) to show the real irradiances.

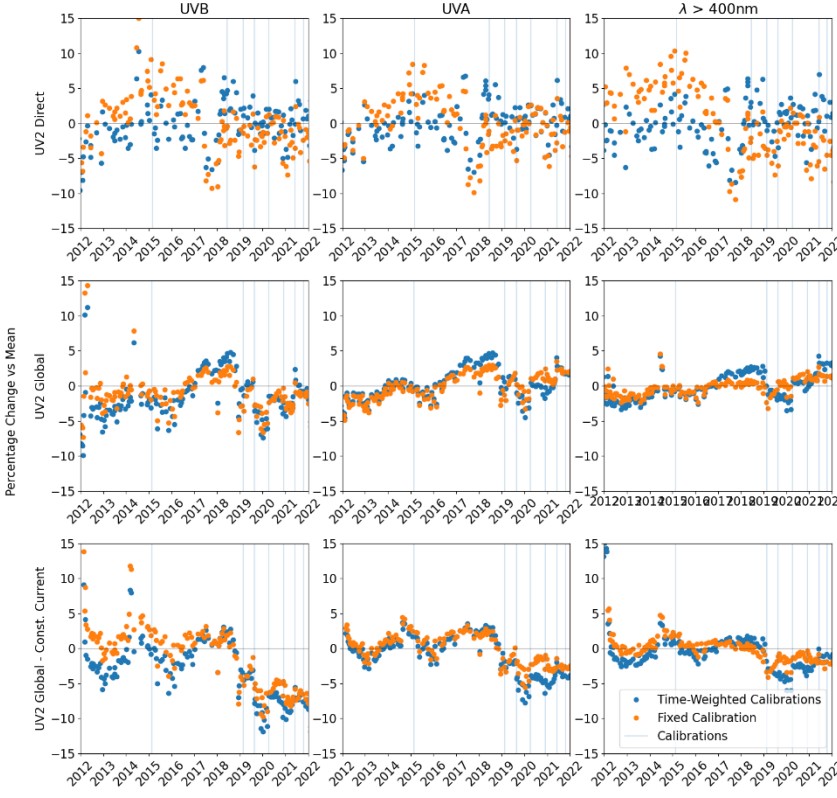


Figure 1: Constant intensity stability measurements for global and direct in the UVB, UVA and for wavelengths greater than 400 nm. Results using a time weighted mean of 1000 W calibrations are shown in blue, and fixed calibration, where the

original voltage response is used, in orange. Blue lines indicate when 1000 W calibrations were available and used in the time weighted mean. The bottom row shows the constant current measurements for UV2 Global that are made immediately after the constant intensity measurements.

For the direct measurements (upper panels), across the three bands, the instrument remained within 5% of the mean value over the entire period, albeit with a high degree of noise. There is also a clear feature in 2017 where the observed irradiance dropped before returning to normal. The trend through time is consistent across the wavelength regions, indicating that the variability is principally due to a changing throughput, with little spectral dependence. This change is therefore likely caused by a change in the instrument throughput from a shift in the fibre position. As expected, using the varying sensitivity reduces the magnitude of the differences from the mean compared to the fixed sensitivity case; the 1000W calibrations have, in part, corrected the instrumental drift. The standard deviation of the of the measurements has reduced from 0.083 $\mu$W cm$^{-2}$ to 0.068 $\mu$W cm$^{-2}$ for UVB and from 1.25 $\mu$W cm$^{-2}$ to 1.03 $\mu$W cm$^{-2}$ for UVA, while the mean is largely consistent (2.1 $\mu$W cm$^{-2}$ to 2.12 $\mu$W cm$^{-2}$ for UVB and 37.13 $\mu$W cm$^{-2}$ to 37.30 $\mu$W cm$^{-2}$ for UVA). Due to the lack of calibrations before 2018, the 2017 discrepancy cannot be objectively corrected.

Results for the global irradiance are shown in the lower panels of Figure 1. Again, across all bands the instrument is stable to within 5%, and within 3% outside the UVB region. The 2017 feature is not present in the global irradiances, providing further evidence that it was caused by the direct fibre throughput. The results exhibit far less noise than in the direct beam irradiances, which shows that the global stability measurements are far more repeatable and therefore unlikely to suffer from the fibre coupling issues that beset the direct observations. There is a discernible drift in the instrument between 2017 and 2019, but it remains within 3% of its mean value. There also appears to be a minor spectral dependence, with the UVB and longer wavelengths both showing more variability and higher differences than UVA between the two measurement modes (time-weighted and fixed calibrations). The lack of difference between the varying and fixed sensitivity results, for the UVA at least, shows that the calibrated sensitivity has not changed significantly, hinting that the variations observed over time are potentially from the variations in the 45 W lamp apparatus. These changes could include filament deposition or erosion, feedback diode position, and performance changes. The constant current measurements, shown in the bottom row, are used to assess lamp performance, the differences between the constant current and intensity are caused by a change in the lamp performance over time. The largest difference is seen for UVB, particularly from 2018 onwards, where values are up to 10% below the mean, however the corresponding constant intensity measurements are within 5%. These results show that the longer-term stability of the lamp is adequately accounted for by using the constant intensity mode, though there are still several features in both datasets that indicate other changes in the apparatus, however, they remain within 5% of the mean. The global results are comparable to those of UV4, however significantly more corrective action and maintenance was undertaken on UV4 producing a much more stable and self-consistent time series.

The mercury lamp measurements are used to determine the wavelength alignment of the spectrometer over time, which is then incorporated into the data processing. As a consistency check, we also determine the stability of the observed instrument line shape. For selected mercury lines, we fit a Gaussian function using a window of 5 nm. A relative shift in the line position is then calculated, with the 296.73 nm line as a reference. The result of this analysis is shown in Figure 2.

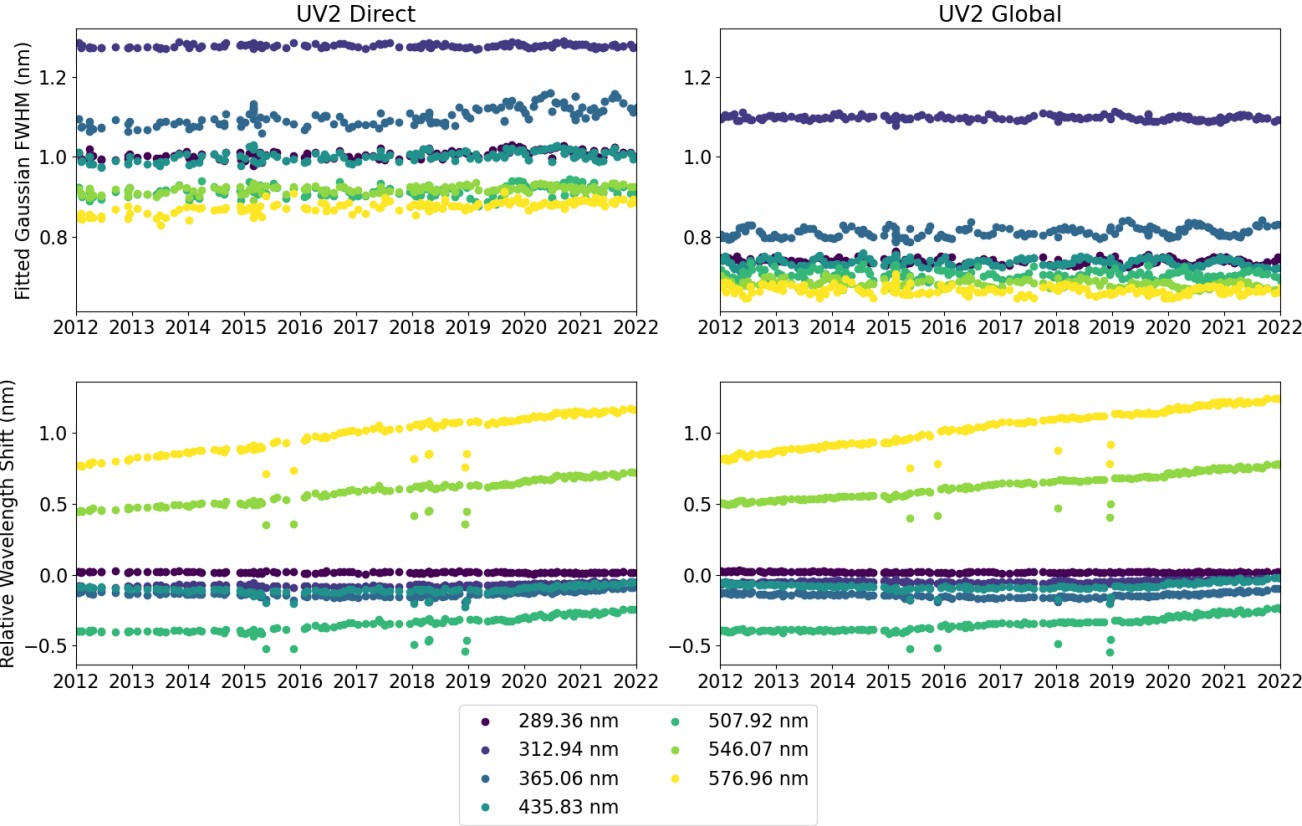

Figure 2: Fitted gaussian FWHM and wavelength shift relative to the mercury line at 296.73m for the direct and global measurements.

The anomalously broader linewidth for the 312.94 'line' arises because it is actually a composite of several unresolved lines. The line near 365 nm is also a doublet. For the direct and global measurements, the full width at half maximum (FWHM) of the fitted slit function is consistent over time for all wavelengths. The direct FWHMs are larger than for the global, indicating a larger apparent effective entrance-slit width for the direct, caused by the lack of a diffuser in the spectrometer entrance aperture for the direct. Drift in the wavelength shift is observed and it is consistent between the global and direct. This shows that there is no spectral dependence caused by the difference in apertures. The wavelength shift highlights the functional wavelength deviation of the instrument, showing increasing shifts with increasing wavelength. The wavelength

shift is increasing with time, particularly for the longer wavelengths, indicating that the monochromator drive system is wearing. However, this is corrected for in real observations by applying a time weighted average of the observed shifts and

also by correcting for additional wavelength deviations by aligning the spectra to solar absorption lines. The global wavelength shifts are also comparable to those observed by UV4, though that instrument is restricted to wavelengths less than 435.83 nm where these the non-linearities are smaller.

In summary, the instrument has performed well in its 10 years of operation. For the global observations, we can confidently

estimate a radiometric accuracy of better than 5% throughout. For the direct measurements, while the majority of measurements are within 5%, there is additional noise caused by varying throughput and a period of a significantly reduced throughput in 2017 which could cause misleading results. However, these changes appear to affect all wavelengths similarly, meaning that outputs like ozone, which depend on the ratios between wavelengths rather than absolute values, should be unaffected. The wavelength alignment and line shape stability has been excellent throughout, with only the expected ageing

of the drive system causing any changes. Such changes are captured and incorporated into the spectra processing algorithm.

## 5 UV Irradiances

To assess instrumental performance for measurement of the irradiance we compare the UVA and UVB irradiances between UV4 and UV2, shown in Figure 3. UV2 and UV4 have simultaneous measurements with the global diffusers so that the data points shown should correspond in time. Data points are additionally screened for cloud conditions. Only cases where UVA

transmittance is greater than 0.3 are considered, avoiding cloud contamination. Additionally, cases where the standard deviation of the diode mounted by the spectrometer aperture divided by its mean is greater than 0.01 are ignored. Such cases are likely to be partially cloud affected.

The agreement between the two instruments is good, with UV2 biased low by 2.6% in the UVB and 3.4% in UVA. The

standard deviation of the percentage differences is higher for UVB (3.7% compared to 3.3%), where the signal to noise ratio of the spectrometer is lower. There is a clear seasonality to the differences, with negative differences during summer and positive during winter, suggesting either a solar zenith angle or temperature dependence which needs to be further investigated. This could likely be linked to the temperature dependence of PTFE diffusers used in measurements of UV irradiance (McKenzie et al., 2005), particularly as the UV4 diffuser is heated and UV2 unheated, but monitored and

corrected (McKenzie et al., 2005).

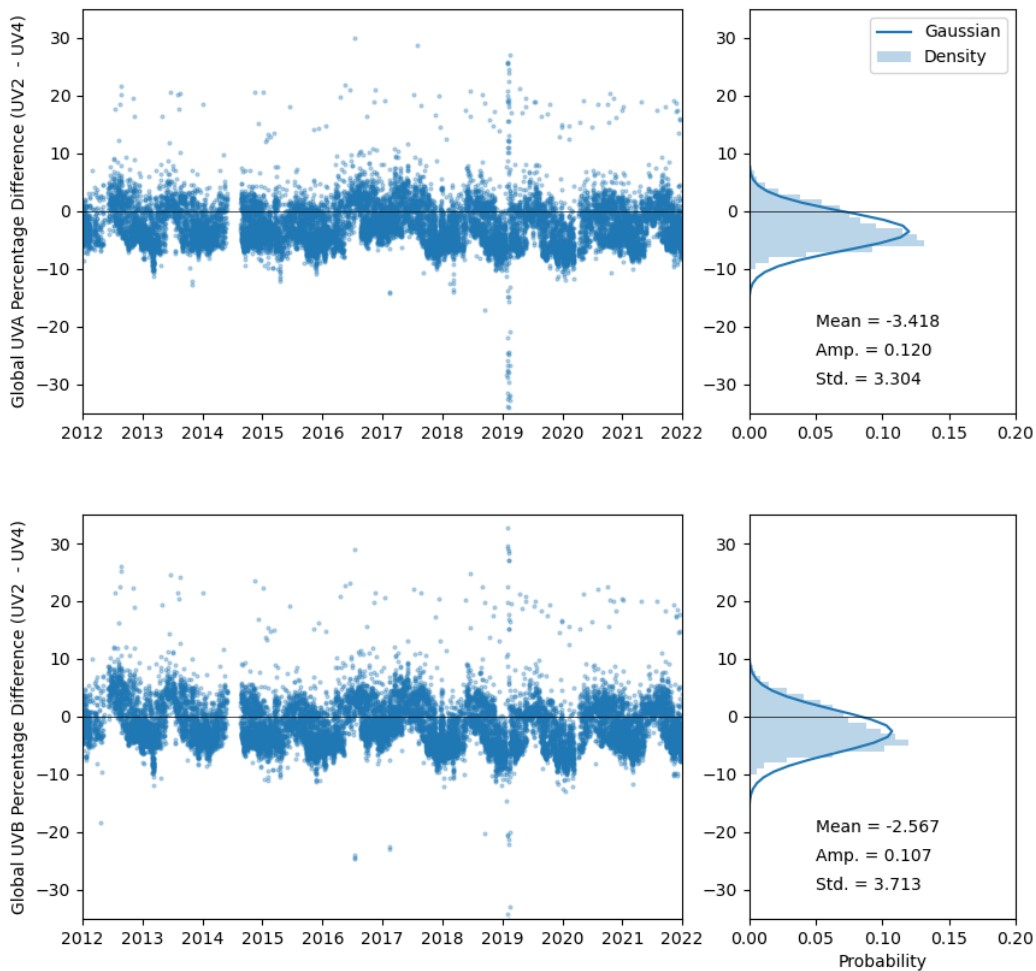

Figure 3: Comparison of the global irradiance measurements of UVA and UVB from UV4 and UV2. Panels on the left show the percentage differences between the instruments with the probability density function shown on the right, with a fitted gaussian and the relevant parameters.

While there is no directly comparable UV comparison available for the direct measurements from UV2, the good agreement between the global measurements of UV4 and UV2, coupled with the results of the stability measurements, gives sufficient confidence to the time series and will be further tested by the ozone and aerosol retrievals.

# 6 Ozone

## 6.1 Methodology

For the direct sun measurements of Ozone by UV2 we use the same approach as in retrievals from Dobson spectrometer, which is an application of Beer's Law. The ozone layer thickness is calculated by comparing the intensity of UV radiation at strong and weakly absorbing wavelengths. In the case of the Dobson, a variable attenuator is used to measure the ratio of the intensities at two wavelengths.

For UV2, the ratio of the intensities is taken from the measured UV spectrum. To remove any wavelength dependencies introduced by unknown aerosol extinctions, two pairs of wavelengths are used with the resulting formula for wavelengths pairs i and j for total column ozone, X, being;

$$X_{ij} = \frac{(N_i - N_j) - [(\beta - \beta')_i - (\beta - \beta')_j]\frac{mp}{p_0}}{[(\alpha - \alpha')_i - (\alpha - \alpha')_j]\mu}$$

Where;

$$N = \log_{10}\left(\frac{I_0}{I_0'}\right) - \log_{10}\left(\frac{I}{I'}\right)$$

For the wavelengths pairs i and j, with I and I' being the measured intensity at the ground at the short and long wavelengths respectively of the wavelength pair. $I_0$ and $I_0$' are the intensities of the two wavelengths but for the solar radiation outside of the atmosphere, calculated from convolving the slit function of the instrument with a high resolution extra-terrestrial solar spectrum (Chance & Kurucz, 2010). Here $\alpha - \alpha'$ and $\beta - \beta'$ are the differences in the ozone absorption coefficient and the Rayleigh scattering coefficient between the wavelength pairs. The airmass factor (m) is the ratio of the actual and vertical paths of the solar radiation and $\mu$ is the ratio of the actual and vertical paths of the solar radiation through the ozone layer assuming an ozone layer altitude of 22 km and site altitude of 0.370 km. The mean station pressure (p) is 970mb and the mean sea-level surface pressure, $p_0$ is 1013 mb. The wavelengths pairs used at Lauder are the AD pairs and the CD pairs, the CD pair applying when $\mu$ is greater than 3.0.

| Wavelength Pair | Wavelengths (nm) | $\alpha - \alpha', [(atm - cm)^{-1}]$ | $\beta - \beta', [atm^{-1}]$ |
|---|---|---|---|
| A | 305.5 – 325.0 | 1.806 | 0.114 |
| C | 311.5 – 332.4 | 0.833 | 0.109 |

| | | | |
|---|---|---|---|
| D | 317.5 – 339.9 | 0.374 | 0.104 |
| AD | | 1.401 | 0.0049 |
| CD | | 0.467 | 0.0071 |

Table 1: α and β values for the individual wavelength pairs along with the combined values (i.e. $(\alpha - \alpha')_i - (\alpha - \alpha')_j$ ) for
the AD and CD pairs used in this study.

The ozone absorption coefficients used for UV2 are the same as those given in (Komhyr, 1980) and the Dobson slit functions, parameterised as trapezoids (Komhyr, 1980) are also applied to the UV2 Spectra to ensure consistency between the two instruments. This means that the UV2 retrievals of ozone will still be susceptible to the known stratospheric
temperature issues, as is the case for normal Dobson retrievals (Bernhard et al., 2005; Brinksma et al., 2000; Komhyr, 1993; Van Roozendael et al., 1998). In short, the Dobson method assumes an ozone cross section at a fixed temperature (229K) whereas the stratospheric temperature varies throughout the year, which can potentially introduce a false seasonality to observations. This method, used by the Dobson yields a precision of 1% and an accuracy of 5% (Basher, 1985). Methods to retrieve ozone from UV spectra have been successfully demonstrated, which can account for these shortcomings, as in (Egli
et al., 2023; Zuber et al., 2021), and will be the subject for future analyses. In terms of validating UV2's performance however, if we are consistent in the assumptions made, an intercomparison is possible and, given the Dobson's high degree of traceability and heritage, is an appropriate path to take.

Ozone can also be calculated using the global irradiances from the UV spectrometers from the ratio of 305 nm to 340 nm
irradiance (Stamnes et al., 1991). In this method, a table of ratios as a function of SZA and ozone is precomputed using the TUV radiative transfer model and measured ratios are compared to find the best matching ozone value with those ratios. A limitation of the method is that it typically assumes a fixed surface albedo and aerosol load. Departures from those can lead to systematic errors.

**6.2 Results**

Here we compare the times series of the direct Dobson, UV4 (global) and the global and direct observations from UV2 which are filtered using the same criteria as in Section 5. The averages of these measurements are found for observations within ± 15 minutes of all direct sun Dobson measurements. The resulting differences are shown in Figure 4.

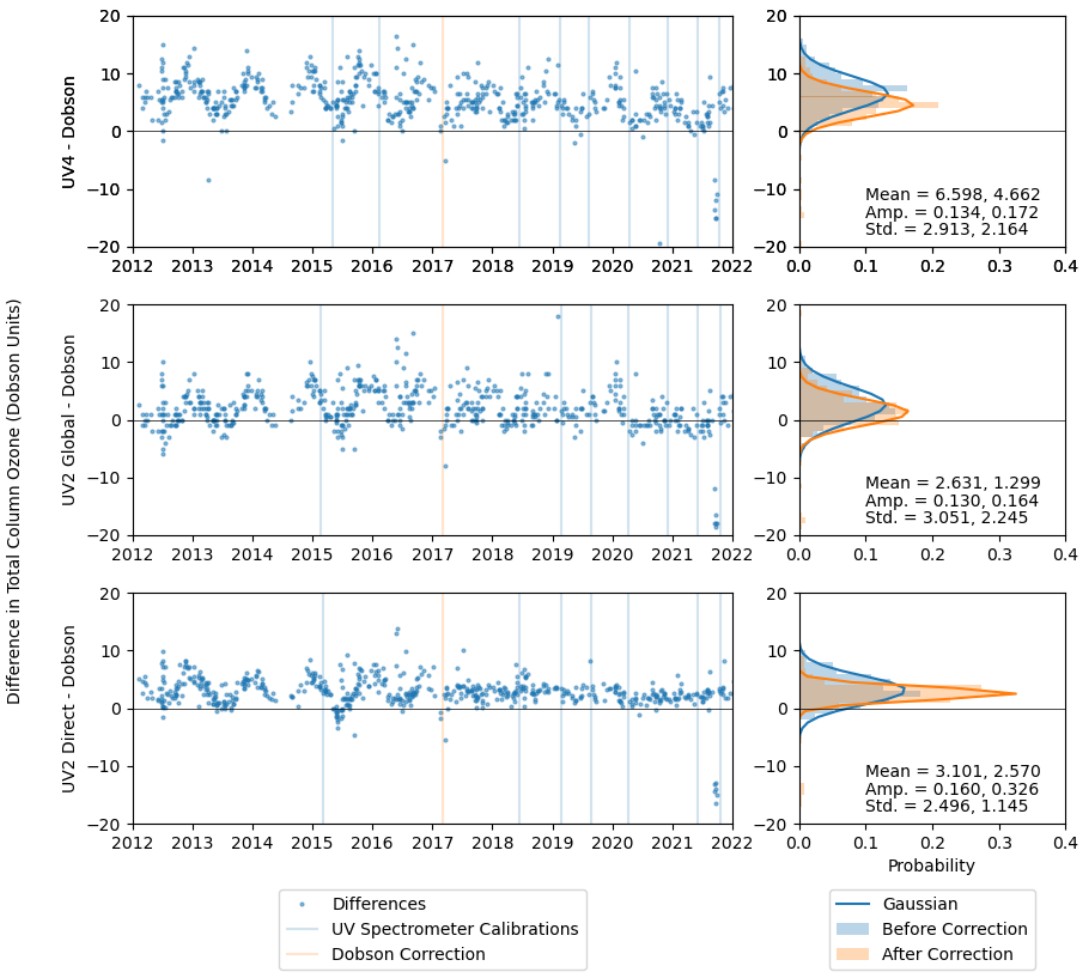

Figure 4: Differences with respect to the Dobson in total column ozone for UV4 and both UV2 measurements. Calibration dates of the UV spectrometers are shown by blue lines and the Dobson intercomparison corrections in orange. The panels on the right show the distribution of differences before the Dobson was corrected (blue) and after (orange) along with the mean, median and standard deviation of the fitted Gaussian respectively.

As can be seen, there is significant variation for all instruments prior to 2017, indicating a seasonal variation in the Dobson data. In February of 2017, the Dobson was compared with the regional standard and correction factors were applied, thereafter the comparisons improve significantly. This seasonal difference was likely caused by stray light in the Dobson which was corrected in 2017. The two global measurements, which utilise lookup tables, perform well, with a mean bias of

4.7 and 1.3 Dobson Units (DU) for UV4 and UV2 respectively, both with a standard deviation of approximately 2.2 DU, the difference in mean bias driven by differences in calibration As expected, the UV2 direct observations are excellent, while there is a modest mean bias of 2.6 DU, the standard deviation has improved to less than 1.2 DU. The increase in mean bias between the global and direct is likely due to differences in calibration and throughput between the global and direct. The Dobson method used by UV2 direct does not strongly rely on the radiometric calibrations by using wide Dobson slit functions, and therefore does not suffer as much from the lack of calibrations discussed earlier. This can be seen for the period prior to the mid 2018 calibration and after the Dobson intercomparison in early 2017, where the differences are consistent with 2018 onwards.

Consistent agreement with Dobson values to within its measurement uncertainties suggests that the spectrometer system could eventually supersede the need to maintain measurements with the Dobson system. This would have huge implications for lowering operational costs and improving data frequency. However, a further, more rigorous analysis would be needed to assess this potential, which is out of scope of this study.

## 7 Aerosol

### 7.1 Methodology

Aerosol optical depth is also calculated using Beer's Law with direct sun observations;

$$I = I_0 e^{-\tau}$$

A bottom-of the-atmosphere reference spectrum of direct irradiance ($I_0$) is computed using TUV with zero aerosol and a fixed ozone amount of 300 DU for each direct sun measurement of UV2, using the solar zenith angle corresponding to the midpoint of the measurement, which is between the forward and reverse scans. The fixed ozone amount is so that it compares directly with the Prede Sky radiometer. By combining the observed spectra (I) we can deduce the aerosol optical depth ($\tau$). Multiple scattering is ignored in this approximate calculation; while important, in the low aerosol case at Lauder and by using the direct beam, the impact of multiple scattering is minimised This was done for the AOD bands of the Sky radiometer and SP02 (400 nm, 412 nm and 500 nm).

### 7.2 Results

Observations of aerosol optical depth by Prede and SP02, both at 1-minute intervals, were interpolated to the times of observations by UV2; the mean of AOD observations within ± 15 minutes of a UV2 measurement. The SP02 AOD

algorithm, from the Australian Bureau of Meteorology, inherently filters out cloud-obstructed data, and by restricting to these times we have screened for clouds across all datasets. Similarly, the preliminary dataset provided by CEReS has also been cloud filtered. The UV2 direct observations are filtered in the same way as for ozone observations. The aerosol optical depth was then calculated for wavelengths of 400 nm and 500 nm for comparison to the Prede, and 412 nm and 500 nm for the SP02. The median AOD measured by the Prede was 0.05 and 0.04 for 412 nm and 500 nm respectively and 0.043 and 0.034 for 412 nm and 500 nm from the SP02, highlighting the low AOD at Lauder and the previously stated challenge in accurately measuring AOD. The differences at 400 nm and 412 nm are shown in Figure 5 and for 500 nm in Figure 6, with the probability density function (PDF) of the differences and the parameters of a Gaussian fitted to the PDF also shown.

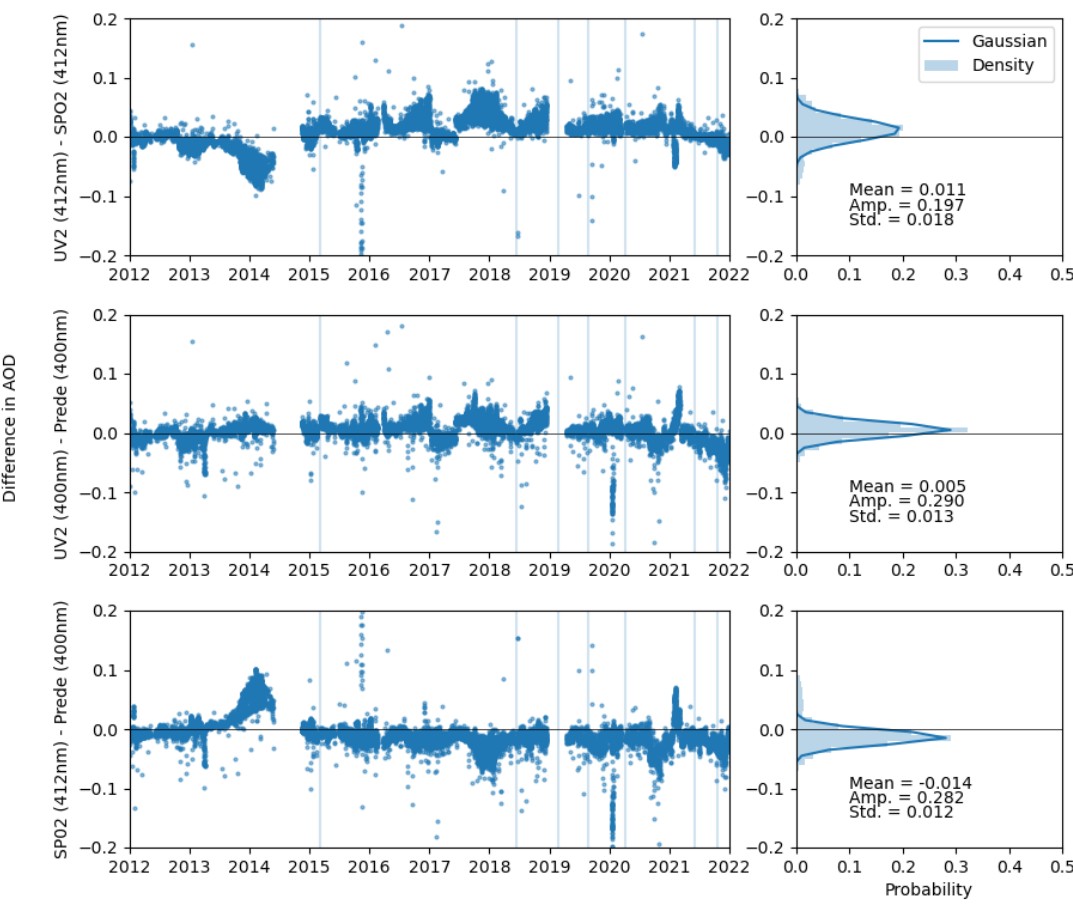

Figure 5: Differences in the aerosol optical depth between UV2 at 400 nm and 412 nm and SP02 at 412 nm and the Prede at 400 nm. Panels on the right show the distribution of differences and a fitted Gaussian with its associated parameters.

At 400-412 nm the agreement between all three instruments is favourable. UV2 is shown to agree with SP02 and the Prede to a comparable accuracy and precisions as SP02 and the Prede agree with each other. In all cases, the mean bias is less than 0.015 and the standard deviation less than 0.02. There are however several pronounced features in the three-time series. At 412 nm, during 2014 there is a marked difference of 0.05 AOD, with UV2 significantly underestimating AOD with respect to the SP02. This is not present in the 500 nm AOD, and on review it is apparent from the diurnal variation on clear days that the 412 nm AOD data are faulty from October 2013 until June 2014, when the SP02 sensor was sent to the Bureau for recalibration and replacement of the damaged 412 nm filter. The discrepancy with UV2 is corrected by 2015, when another full calibration of UV2 was made. There are also several spikes in the Prede data, notably early 2020, that are yet to be resolved. They are however short lived, suggesting that they are a result of dust build up before being cleaned, and do not affect the comparison significantly. UV2 also appears to have improved following regular calibrations from 2018 with a reduction in prolonged periods of differences such as those seen from 2015 to mid-2018.

Much of the same is true when we look at 500 nm, however the comparisons to UV2 aren't quite as favourable when compared to the differences between SP02 and the Prede. There the mean of the Gaussian PDF is -0.006 with a standard deviation of 0.007. Further work is needed to improve the calibration of UV2 at longer wavelengths, where, as shown previously, the impact of shifts in wavelength alignment is felt most strongly. In addition, absorption by ozone is present in this region and may not be adequately corrected.

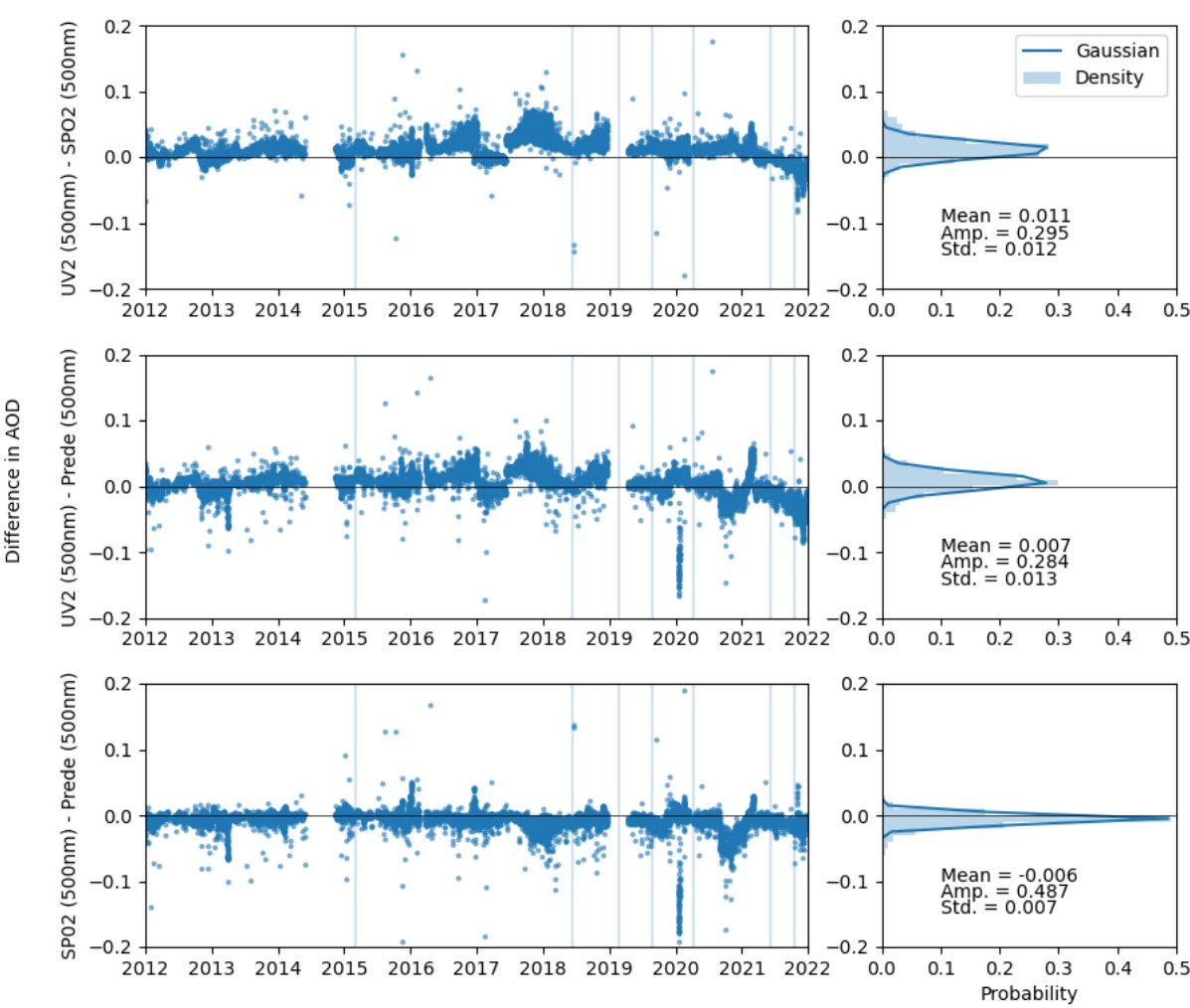

Figure 6: Differences in the daily mean aerosol optical depth between UV2, Prede and SP02 at 500 nm. Panels on the right show the distribution of differences and relevant statistics.

360

## 8 Summary

We have demonstrated the utility of the UV2 instrument with its capability of near-simultaneous direct and global UV irradiance measurements. Over the past 10 years it has proven stable to within 3% and 5% for measurements of global and direct irradiances respectively with only minor issues with the direct measurements in 2017-2018, which have since been corrected. The global irradiances measured by UV2 compared well with those measured by the NDACC certified UV4, with mean biases of 2.4% and 3.3% for UVB and UVA respectively, with standard deviations less than 4%. With the direct measurements we were able to measure ozone to well within 6 DU (~2%) of the reference Dobson, which compares well with more recent developments, such as those from the Brewer (Up to 5% differences from Dobson,(Gröbner et al., 2021)), Koherent system (1.64%,(Egli et al., 2023)) and QASUME (1%, (Egli et al., 2022)), although further work would be needed to provide direct comparison, such as investigating the use of a spectral fitting. This is an ideal complement to the Dobson time series as it is fully automated, so providing several estimates of ozone throughout daylight hours, and thus allowing for data gaps to be filled. Aerosol optical depth was also computed and compared to the SP02 and Prede observations, with UV2 comparing well to both, with the difference in measured optical depths typically within 0.015. It should be noted however, that the AOD is extremely low at Lauder, with values above 0.1 being exceptionally rare. As such, uncertainties will approach the detection limit of the instrument. Again, we have shown that UV2 is a complementary measurement system. It can extend the measurements of AOD to cover various wavelengths and can be used as an intermediate comparison between the Prede and SP02 based measurements. Further work is needed to improve the agreement at 500 nm, where wavelength alignment and ozone effects will be present. Measurements will continue at Lauder and longer-term tests will be made to determine whether the instrument can be used to supersede Dobson and aerosol radiometer measurements. In addition, the utility combining global and direct measurements has yet to be fully explored and will be a topic of future research now that the instrument performance has been characterised.

## 9 Acknowledgments

This work has been funded by the New Zealand Ministry of Business, Innovation and Employment (MBIE). We acknowledge the support of CEReS in particular Hitoshi Irie, for their support with data from the Prede Sky Radiometer. We are also grateful to Bruce Forgan who, as part of the Australian Bureau of Meteorology (BoM) was instrumental in the development of aerosol measurements from the SP02 system.

## Data Availability

UV2 and SP02 data is available from authors on request. Dobson and UV4 data are available from the NDACC and the Prede Skyradiometer data is available from http://atmos3.cr.chiba-u.jp/skynet/lauder/lauder.html.

## Author Contributions

Alex Geddes was responsible for the analysis of UV2 spectra and intercomparisons with other datasets and was the primary author of the paper. Ben Liley and Richard McKenzie supported the writing and analysis. Michael Kotkamp maintained the instruments used in the study. Richard Querel was responsible for the Dobson dataset and supported the writing effort.

## Competing Interests

The authors declare no financial interest beyond their benefits from employment.

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
