# Peer review of "Novel use of an adapted UV Double Monochromator for measurements of global and direct irradiance, ozone and aerosol"

_Atmospheric Measurement Techniques, 2023_

## Referee Comment (RC2)

**General Comments**

The paper shows good scientific significance by promising results of an adapted established measurement system extended to total ozone columns and aerosol optical depth. There are some interesting features in the instrument, like the beam splitting and switching between two entrance optics and the fact that established double monochromator systems might be extended with such a setup. Furthermore, the time period presented of the measurements itself is of interest to be published.

The presentation quality is good but lacks a bit in the introduction part in not mentioning other recently developed new instruments for total ozone column and aerosol optical depth or other double monochromator systems already used for these measurements. This would be worth adding to connect it better to the scientific community, see later specific comments. In addition, the good scientific quality could benefit by extending the uncertainty part and the comparison to established systems and their measurement uncertainties.

I recommend to publish this paper at AMT but would recommend to add, even if just short, some of the specific comments mentioned.

**Specific Comments**

- Line 1: Maybe the brand of the double monochromator is not important in the title since also other double monochromators might be suited. Would be in the abstract sufficient or even in the measurement system description. Maybe instead of only the "use" the "adaption" is worth mentioning. Novel use of an adapted UV Double Monochromator for measurements of global and direct irradiance, ozone and aerosol.
- Around line 50: In order to connect the paper better to the scientific community it would be worth citing also other recent developments for AOD and TOC like, e.g.: https://doi.org/10.5194/amt-16-2889-2023, https://doi.org/10.5194/amt-15-1917-2022, https://doi.org/10.5194/amt-11-2477-2018, https://doi.org/10.1016/j.atmosenv.2018.02.036, https://doi.org/10.5194/amt-14-4915-2021
- Line 66: Is it possible to state the accuracy of this switch or in other words the reproducibility accuracy of this optical switching? Long term effects of the switching? Would also fit in chapter 4. This is of interest for readers since it seems a moving and maybe critical part which was, I guess, checked and optimized. Maybe even developed. A little bit more information would be beneficial also for the scientific community if somebody else would like to extend his double monochromator system and reproduce the science.
- Line 72: The reader might ask himself what kind of performance assessment this is?
- Chapter 2: Please state the measurement interval for TOC and AOD measurements and the scanning time. I guess as well 15 min all 15 min in the same SZA/airmass range?
- Line 101: Please state the traceability of this 45 W lamp to primary standard since it seems to be a working standard? Maybe state uncertainty wavelength dependent.
- Line 106: Please state as well the traceability and calibration uncertainty

- Line 144: Recommendation in order to find out if it is really the 45 W apparatus would be reproduction measurements to quantify this performance changes. Reproduction measurements on a short term in order to separate this from lamp changes. For next time.
- Line 198: In terms of structure Chapter 6. and 6.1 seems a bit strange without having a chapter 6.2. I would recommend to write 6 Ozone as main header and then directly 6.1 Retrieval description 6.2 Results. I would recommend the same for Aerosol and Irradiance to have some consistent structure.
- Line 242: As in chapter 3? Filtering is just there explained and not in chapter 6 as far as I have seen.
- Chapter 6: Since a full spectrum is measured also full spectral retrieval or spectral band retrievals could be used. Maybe cite other full spectrum or spectral band-based retrieval methods, see publications above. Maybe worth considering these methods for the future and compare them with the one used.
- Line 252: Beside calibration also a seasonal drift correction was applied? Please state clearer and cite reference method if available.
- Chapter 6: Can you discuss why the direct method is showing a higher offset compared to the global method?
- Chapter 6: Can you state the absolute measurement uncertainty of the Dobson reference used? A comparison of the absolute measurement uncertainty of the new system would be interesting. Also, SZA/air mass dependent. Maybe too much for this paper but worth considering. See publications like: https://doi.org/10.5194/amt-11-3595-2018
- Chapter 7: Some citations would be beneficial. There is a very new publication which might be helpful, see: https://doi.org/10.5194/amt-2023-105. It also gives measurement uncertainties, etc. which could be compared.
- Line 311: Please make clear what this 6 DU are. Difference measured to reference Dobson?
- Chapter 8: UV irradiance results are not summarized. Maybe worth a sentence.
- Chapter 8: In general, it would be good to compare the measurement accuracy or at least the deviation to reference instruments to other existing or rather newly developed systems such as array spectroradiometer-based devices or other double monochromator-based systems. See above mentioned publications. This would improve the scientific classification enormously and would connect it better to the scientific community.

**Technical Corrections**

- Line 70: I would recommend to delete the word "useful" since this is up the reader.
- Figure 3: Please use a finer scaling on y-axis. 20% increment is very rough and hard to read. Suggested +/- 35% instead of ~50% with 5% increment or at least 10% increment.
- Line 243: Typo, just one +- needed.
- Line 276: Typo, +/- instead of +=

---

## Author Response (AR1)

**Reviewer 1 Comments and Corrections**

- The paper discusses the performance of a spectroradiometer that has been in operation for about 10 years in Lauder, New Zealand, and presents as additional products total ozone and aerosol optical depth retrievals and comparison to other collocated standard instruments. The quality of the presentation is satisfactory, although it could have been further improved with more careful and detailed discussion of the different topics. In general the methodologies are mainly outlined instead of being presented in detail. Nevertheless, the paper is a useful documentation of the system and its performance and for this reason it can be accepted for publication in AMT as long as the specific comments below are addressed.

Overall Comment: We thank the reviewer for taking the time to review this paper and their thoughtful comments. We have addressed all of the comments below and have added in a substantial amount of extra detail and information regarding the methods used. The reviewers comments have helped to significantly enhance and improve the paper.

- The introduction is severely biased to local studies neglecting the significant relevant work on spectral UV monitoring done in several other locations worldwide.

As mentioned above, several references have been added to aid in placing the instrument in the global context.

- In all figures showing differences, it would help to draw a line at 0 difference to guide the eye of the reader.

This has been added and enhances the readability signficantly

- 65: how you cope with the time difference between global and direct spectra (and consequent change in solar zenith angle) to calculate the diffuse irradiance spectra?

Time difference will be 15 minutes, during stable conditions, this is enough to approximate the diffuse irradiance. Text has been updated to clarify this.

- 69: how long it takes to acquire one full spectrum?

Updated the text to include this information, 4.5 minutes using a forward and reverse scan pair

- 70: 0.65 nm is probably the "average" resolution (according to Figure 2)

Correct, Clarified in the text

- 87: To which instrument refers the "accuracy of the spectral calibration"? Unless it is meant "Langley calibration".

Yes, Langley, amended the text

- 101-103: Is there any reference to provide more details on this method? How the 45W lamps are used/mounted when measuring the direct irradiance port?

Addressed in the text with a more thorough description of the setup

- 114: As you say in line 158, there is no diffuser for the direct port. Therefore the "global and direct diffuser" should probably be replaced by "global and direct ports".

Good point, amended in text

- 114: Figure 1 refers to the constant intensity mode which is based on measurements with a UVA photodiode. It would be interesting to discuss the stability when the constant current mode is used.

This is used to assess the long term performance of the 45W lamp and a short discussion has been included along with extra panels in figure 1.

- 128-129: How it is ensured that the change in 2017 (and other shorter-term changes) are not caused by the 45 W lamps? How the stability of these lamps with time is ensured?

Constant current measurements are used to assess lamp performance over time, however other short term changes to the apparatus are still possible.

- 129-130: Please include some statistics for the two cases (fixed and varying sensitivity), e.g. mean and standard deviation of the differences.

The mean and stand deviation of the UVA and UVB measurements have been included in the text.

- 161: Please elaborate on why you think that the wavelength drift indicates a linearity problem. And if such problem exists, how this could affect the actual spectra that have a large dynamic range?

The magnitude of the wavelength shift increases with wavelength. The actual spectra have the corrections from the wavelength shift measurements applied to them to mitigate this. It also uses solar absorption lines to finely adjust the spectra. This has been added to the text.

- 165: How the performance of this instrument compares to the performance of the UV4 system? Are the observed changes in sensitivity and wavelength stability within the norms for an NDACC affiliated instrument?

Comparable to UV4 with two caveats, firstly UV4 does not extend to the longer wavelengths so comparison is limited, however within the lower wavelengths, the magnitude of shifts is similar. The sensitivity shown in the constant intensity plots is also of a similar magnitude, however the UV4 instrument was maintained to a higher standard, resulting in a smoother timeseries. These comments have been added to the text

- 177: What are the diffuser diode and the mean diode?

Good point, an oversight on our part. A description has been added to the instrument section. A diode is mounted to the aperture of the spectrometer to monitor changes in the incoming radiation, primarily driven by clouds moving across the solar disc.

- 215: Have the ratios of the extraterrestrial intensities been calculated by Langley plots? And if yes, how often these have been done?

No, extra terrestrial intensities are calculated using the TUV code. Clarified in the text

- 223: If I have understood correctly, UV2 scans in steps of 0.2 nm. Then how the wavelength with decimal 5 or 9 in table 1 are achieved?

Weighted slit functions are applied to the central wavelengths in table 1

- 263: I disagree with this statement. The replacement of a total ozone instrument at a particular station requires a though intercomparison and assessment a any differences and limitations. If can certainly no be based on a simple plot of differences. See for example, Groebner et al., 2021 10.5194/amt-14-3319-2021.

We completely agree with the reviewer, we have amended the text to more strongly state that this study only shows that UV2 'could' potentially supersede the Dobson, but a further, more rigorous analysis would be needed, which is beyond the scope of this paper

- 268: Add "of direct irradiance" after "reference spectrum". Has this reference spectrum calculated for the specific solar zenith angle of each individual spectral measurement (400, 412 and 500 nm)?

The reference spectrum is calculated for the sza of the midpoint of the measurement, given that each measurement takes 4.5 minutes and is the average of a forward and reverse scan (with the midpoint at turn around) it is a reasonable to neglect the variation in SZA throughout the measurement as it would roughly average out. This has been clarified in the text in the measurement strategy.

- 270: Why you have to convert the direct irradiance to a horizontal surface? In fact to deduce the aerosol extinction you need the actual path of radiation. In addition, please elaborate a bit more on the methodology, so that inexperience readers can understand what you are doing.

Clarified in text, TUV outputs direct irradiance on a horizontal surface so needed to be corrected to the direct path. The formula used has also been stated to aid readability

- 275: 1-minute data cannot be interpolated to the times of observations by UV2 which have much coarser resolution.

Correct, poor wording on our part, should be filtered and averaged, this is reflected in the text

- 309: Please add after "proven stable", "to within XX% for global and YY % for direct irradiance".

This is an important point to include in the summary section and has been added to the text

Technical:

- 64: add "radiation" after "receives"

Done

- 69: replace full stop with comma at the end of the line

Done

- 262: delete "the" after "its"

Done

- 276: replace "+=" with "±"

Done

**Reviewer 2 Comments and Corrections**

- General Comments The paper shows good scientific significance by promising results of an adapted established measurement system extended to total ozone columns and aerosol optical depth. There are some interesting features in the instrument, like the beam splitting and switching between two entrance optics and the fact that established double monochromator systems might be extended with such a setup. Furthermore, the time period presented of the measurements itself is of interest to be published. The presentation quality is good but lacks a bit in the introduction part in not mentioning other recently developed new instruments for total ozone column and aerosol optical depth or other double monochromator systems already used for these measurements. This would be worth adding to connect it better to the scientific community, see later specific comments. In addition, the good scientific quality could benefit by extending the uncertainty part and the comparison to established systems and their measurement uncertainties. I recommend to publish this paper at AMT but would recommend to add, even if just short, some of the specific comments mentioned. Specific Comments

Overall Comment: We thank the reviewer for taking the time to review this paper and their thoughtful comments. We have addressed all of the comments below and have added in a substantial amount of extra detail and information regarding the methods used. In addition we have added extra references and details to better place the system in a global context as suggested by the reviewer. The reviewers comments have helped to significantly enhance and improve the paper and are very much appreciated.

• Line 1: Maybe the brand of the double monochromator is not important in the title since also other double monochromators might be suited. Would be in the abstract sufficient or even in the measurement system description. Maybe instead of only the "use" the "adaption" is worth

mentioning. Novel use of an adapted UV Double Monochromator for measurements of global and direct irradiance, ozone and aerosol.

Agreed, title changed

• Around line 50: In order to connect the paper better to the scientific community it would be worth citing also other recent developments for AOD and TOC like, e.g.:

https://doi.org/10.5194/amt-16-2889-2023,

https://doi.org/10.5194/amt-15-1917-2022,

 https://doi.org/10.5194/amt-11-2477-2018

https://doi.org/10.1016/j.atmosenv.2018.02.036,

https://doi.org/10.5194/amt-14-4915- 2021

Thanks for the suggestion. Several additional citations, including the above have been added to the text to better provide context for the system.

 • Line 66: Is it possible to state the accuracy of this switch or in other words the reproducibility accuracy of this optical switching? Long term effects of the switching? Would also fit in chapter 4. This is of interest for readers since it seems a moving and maybe critical part which was, I guess, checked and optimized. Maybe even developed. A little bit more information would be beneficial also for the scientific community if somebody else would like to extend his double monochromator system and reproduce the science.

The short term repeatability of this is better than 1% but over the longer term corrections will need to be applied through regular calibration. This has been added to the text.

• Line 72: The reader might ask himself what kind of performance assessment this is?

Clarified in the text, it is two assess performance at longer wavelengths compare to UVA and UVB

 • Chapter 2: Please state the measurement interval for TOC and AOD measurements and the scanning time. I guess as well 15 min all 15 min in the same SZA/airmass range?

Clarified in text. 5 minute measurements

 • Line 101: Please state the traceability of this 45 W lamp to primary standard since it seems to be a working standard? Maybe state uncertainty wavelength dependent.

This has been addressed in the text, the 45W lamp can be validated against NIST traceable QTH lamps which are the primary means of calibration, provides an additional uncertainty of 1-2%

• Line 106: Please state as well the traceability and calibration uncertainty

Addressed in text, traceable to NIST standards with an uncertainty of 3%.

 • Line 144: Recommendation in order to find out if it is really the 45 W apparatus would be reproduction measurements to quantify this performance changes. Reproduction measurements on a short term in order to separate this from lamp changes. For next time.

We have added in additional figures and commentary on the use of constant current measurements to assess long term performance of the 45W lamps. These show that the constant intensity mode is adequately constraining the variability to less than 5%. Repeated measurements in the short term would indeed be useful to investigate the sub 5% variations and might be something we investigate further.

• Line 198: In terms of structure Chapter 6. and 6.1 seems a bit strange without having a chapter 6.2. I would recommend to write 6 Ozone as main header and then directly 6.1 Retrieval description 6.2 Results. I would recommend the same for Aerosol and Irradiance to have some consistent structure.

Done

• Line 242: As in chapter 3? Filtering is just there explained and not in chapter 6 as far as I have seen.

As in chapter 5, corrected

• Chapter 6: Since a full spectrum is measured also full spectral retrieval or spectral band retrievals could be used. Maybe cite other full spectrum or spectral band-based retrieval methods, see publications above. Maybe worth considering these methods for the future and compare them with the one used.

A reference to this effect has been added to the text, explaining the potential for this and that this might be a subject of further research. For an initial comparison to the Dobson however, the method used is appropriate

• Line 252: Beside calibration also a seasonal drift correction was applied? Please state clearer and cite reference method if available.

A seasonal drift correction was not applied. The apparent decrease in the seasonal difference from 2017 was is likely due to the stray light correction applied to the Dobson, which if uncorrected would result in a seasonal difference in ozone.

• Chapter 6: Can you discuss why the direct method is showing a higher offset compared to the global method?

Global method relies more strongly on calibrations, in the direct method, wide slit functions are used, minimising the dependence.

• Chapter 6: Can you state the absolute measurement uncertainty of the Dobson reference used? A comparison of the absolute measurement uncertainty of the new system would be interesting. Also, SZA/air mass dependent. Maybe too much for this paper but worth considering. See publications like: https://doi.org/10.5194/amt-11- 3595-2018

The precision of direct sun Dobson measurements is estimated at 1% with a 5% accuracy according to Basher 1985. This has been added to the text.

• Chapter 7: Some citations would be beneficial. There is a very new publication which might be helpful, see: https://doi.org/10.5194/amt-2023-105. It also gives measurement uncertainties, etc. which could be compared.

Thank you for the suggestion, further comparison will be the subject of future work. The paper at present already compares 3 instruments and this is a somewhat preliminary paper assessing the performance of the UV2 instrument.

• Line 311: Please make clear what this 6 DU are. Difference measured to reference Dobson?

Reference Dobson, clarified in text.

• Chapter 8: UV irradiance results are not summarized. Maybe worth a sentence.

Added a sentence summarising the results and comparison to UV4

• Chapter 8: In general, it would be good to compare the measurement accuracy or at least the deviation to reference instruments to other existing or rather newly developed systems such as array spectroradiometer-based devices or other double monochromator-based systems. See above mentioned publications. This would improve the scientific classification enormously and would connect it better to the scientific community.

Comparisons have been included and are useful addition to the discussion, with the caveat that further work is needed to explore them as this is very much a preliminary study of the instrument.

Technical Corrections

Line 70: I would recommend to delete the word "useful" since this is up the reader.

Done

Figure 3: Please use a finer scaling on y-axis. 20% increment is very rough and hard to read. Suggested +/- 35% instead of ~50% with 5% increment or at least 10% increment.

Done

Line 243: Typo, just one +- needed.

Done

Line 276: Typo, +/- instead of +=

Done

---

## Author Response (AR2)

Thank you to both referees for their prompt responses. All corrections have been made and we are grateful for having them spotted by the referees.

**Anonymous Referee #1 – Report 2**

L 54: Replace "and" with coma before "Brewer"

**Done**

L 159: Units are missing in the means

**Done**

L 162: Replace "global diffuser" with "global irradiance"

**Done**

L 257: TUV uses a measured extraterrestrial spectrum to calculate the intensities throughout the atmosphere. I don't see how TUV is used to derive the extraterrestrial intensities. I would expect that you used a measured solar spectrum convoluted with slit function(s) of your instrument.

**Corrected, we convolve the slit function with the solar spectrum from Chance and Kurucz 2010. We have partially incorporated this function into TUV for simplicity of operation**

L 326: How multiple scattering is ignored? You cannot avoid it in the measured spectra and I am sure that in the standard mode of TUV runs multiple scattering is included. Furthermore multiple scattering is not negligible in the direct beam especially when aerosol optical depth is high.

**As stated in the text TUV is ran with zero aerosol load and 300 D.U of ozone, this is to be directly compare with the Prede sky radiometer. In the low aerosol optical depth environment of lauder, negligible multiple scattering is a reasonable approximation to make for a proof of concept calculation. A comment that it is restricted to low AOD cases has been added to the text.**

**Anonymous Referee #2 – Report 1**

54: BTS-2048-UV array spectrometer (typo and write precise, this one is a spectroradiometer) --> BTS2048-UV array spectroradiometer. Pandora is a spectrometer since not all are radiometric calibrated, okay. The Brewer is also a spectroradiometer, however the manufacturer calls it spectrophotometer, so okay.

**Corrected the BTS-2048-UV case**

124: 1-2 percent --> 1% to 2%

**Done**

195: The wavelength shift plots also highlight the non-linearity of the instrument, showing increasing shifts with increasing wavelength (not easy to read) --> The wavelength shift highlights the functional wavelength deviation of the instrument, showing increasing shifts with increasing wavelength.

**Done, agreed, easier to read now**

199: and by correcting for non-linearity by aligning the spectra to solar absorption lines (I would not call it non-linearity since a wavelength calibration is always non-linear to a certain level, it's an error or deviation) --> and by correcting for errors by aligning the spectra to solar absorption lines (also a . (dot) is missing after the sentence).

**Done, changed non-linearity to wavelength deviation.**

335: 400nm and 500nm (sometimes the unit with spaces, sometimes not, please be consistent) --> 400 nm and 500 nm

**Corrected, and all cases of unit without spaces**

338: The differences at 400 and 412 nm (add unit) --> The differences at 400 nm and 412 nm (same in lines 48, 68 and 84. These are very minor ones, but help some readers)

**Corrected, and identified and fixed several other cases**

346: 400-412nm --> 400 nm - 412 nm

**As above**

375: biases of 2.4 and 3.3 for UVB and UVA (missing percent sign?) --> biases of 2.4 % and 3.3 % for UVB and UVA

**Correct, added %**

376: 6 D.U (remove dot) --> 6 DU

**Corrected to DU along with another case of D.U**